# Assessing Feeding Difficulties in Children Presenting with Non-IgE-Mediated Gastrointestinal Food Allergies—A Commonly Reported Problem

**DOI:** 10.3390/nu16111563

**Published:** 2024-05-22

**Authors:** Adriana Chebar-Lozinsky, Claire De Koker, Robert Dziubak, Daniel Lorber Rolnik, Heather Godwin, Gloria Dominguez-Ortega, Ana-Kristina Skrapac, Yara Gholmie, Kate Reeve, Neil Shah, Rosan Meyer

**Affiliations:** 1Department of Allergy and Immune Disorders, Murdoch Children’s Research Institute, Parkville, VIC 3052, Australia; driclk@yahoo.com.br; 2Macassar Community Health Centre, Department of Health and Wellness, Western Cape Government, Elsenburg 7607, South Africa; claire.dekoker@westerncape.gov.za; 3Department of Gastroenterology, Great Ormond Street Hospital for Sick Children, London WC1N 3JH, UK; 4Department of Obstetrics and Gynaecology, Monash University, Melbourne, VIC 3800, Australia; daniel.rolnik@monash.edu; 5Department of Gastroenterology, Universidad Autonoma de Madrid, 28049 Madrid, Spain; gloriadominguezortega@yahoo.es; 6Department Gastroenterology and Nutrition, Nino Jesus Univesity Children Hospital, 28009 Madrid, Spain; 7HCA Healthcare, London W1G 0PU, UK; info@london-nutrition.com (A.-K.S.); neil.shah@hcaconsultant.co.uk (N.S.); 8Department of Nutrition, Simmons University, Boston, MA 02115, USA; yara.gholmie@simmons.edu; 9Royal Hospital for Children and Young People in Edinburgh, Edinburgh EH16 4TJ, UK; reeve_kate@hotmail.com; 10Department Dietetics, University of Winchester, Winchester SO22 4NR, UK; 11Department of Medicine, KU Leuven, 3000 Leuven, Belgium

**Keywords:** non-IgE-mediated food allergy, feeding difficulties, children, food allergy, food refusal

## Abstract

Many guidelines have been published to help diagnose food allergies, which have included feeding difficulties as a presenting symptom (particularly for non-IgE-mediated gastrointestinal allergies). This study aimed to investigate the prevalence of feeding difficulties in children with non-IgE-mediated gastrointestinal allergies and the association of such difficulties with symptoms and food elimination. An observational study was performed at Great Ormond Street Hospital for Children NHS Foundation Trust, London, UK. Children aged 4 weeks to 16 years without non-allergic co-morbidities who improved on an elimination diet using a previously published Likert scale symptom score were included. This study recruited 131 children, and 114 (87%) parents completed the questionnaire on feeding difficulties. Feeding difficulties were present in 61 (53.5%) of the 114 children. The most common feeding difficulties were regular meal refusals (26.9%), extended mealtimes (26.7%), and problems with gagging on textured foods (26.5%). Most children (40/61) had ≥2 reported feeding difficulties, and eight had ≥4. Children with feeding difficulties had higher rates of constipation and vomiting: 60.7% (37/61) vs. 35.8% (19/53), *p* = 0.008 and 63.9% (39/61) vs. 41.5% (22/53), *p* = 0.017, respectively. Logistic regression analysis demonstrated an association between having feeding difficulties, the age of the child, and the initial symptom score. Gender and the number of foods excluded in the elimination diet were not significantly associated with feeding difficulties. This study found that feeding difficulties are common in children with non-IgE-mediated gastrointestinal allergies, but there is a paucity of food allergy specific tools for establishing feeding difficulties, which requires further research in the long-term and consensus in the short term amongst healthcare professions as to which tool is the best for food allergic children.

## 1. Introduction

Food allergy has been described as the “second wave” of the allergy epidemic [1], and in some countries, its prevalence in early childhood has reached 10% [2]. Food allergies can be antibody-mediated [Immunoglobulin E-mediated (IgE)] or cell-mediated, also known non-IgE-mediated allergies, affecting the skin and the respiratory and gastrointestinal tracts [3]. Many guidelines have been published with symptoms to assist in the diagnosis of food allergies and include feeding difficulties as a presenting symptom, in particular in non-IgE-mediated gastrointestinal allergies [4,5,6,7]. Although it is well accepted by healthcare professionals that this is a common presenting symptom, limited prospective studies [8] have been published to substantiate the inclusion of feeding difficulties as part of the diagnostic criteria and to also describe the type of feeding difficulties commonly seen.

A recent systematic review has found that the prevalence of feeding difficulties in both antibody- and cell-mediated food allergies ranged between 13.6% to 40% [9]. More specifically, Meyer et al. [10] published a retrospective study on children with non-IgE-mediated gastrointestinal allergies and found that 30% of the children had feeding difficulties recorded as a problem in their medical notes. Documented presentations of feeding difficulties ranged between publications from refusing to eat, crying during mealtimes, pocketing food in the mouth, gagging on textured foods, slow eating, fear of food, and food specificities [11,12]. As many studies in the food allergy domain are retrospective in nature, this study aimed to investigate the prevalence of feeding difficulties in children with non-IgE-mediated gastrointestinal allergies at diagnosis and after the elimination diet and the association of such difficulties with symptoms and food elimination. 

## 2. Methods

### 2.1. Study Design and Population

This was an observational study performed at the tertiary gastroenterology department of Great Ormond Street Hospital for Children NHS Foundation Trust, London, United Kingdom (UK). All parents of children aged 4 weeks to 16 years without non-allergic co-morbidities (i.e., cerebral palsy, cardiac disorders) who were required to follow an elimination diet for the diagnosis of suspected non-IgE-mediated gastrointestinal food allergies were approached to take part in this study. A Likert scale questionnaire that had previously been published by Lozinsky and is available as a supplement (Appendix A) [13] was administered prior to starting the elimination diet and again at 4 weeks after starting the food elimination. As described in the publication by Lozinsky et al. [13], the questionnaire measured the severity of nine symptoms (diarrhoea, constipation, vomiting, rectal bleeding, abdominal pain, flatus, bloating, screaming/back arching/irritability, food aversion) individually from 0 (no symptom) to 5 (most severe) and collectively from 0 to 45. It was not possible to assign a specific score to measure improvement, as every individual child would exhibit different symptoms at different severity along the non-IgE-mediated food allergy journey. Any improvement in score (meaning lowering in the score) was interpreted as an improvement following the elimination diet. The research team were responsible for administering the questionnaire to all patients. The limitation of using such a score was discussed in by the publication by Lozinsky et al. [13].

### 2.2. Growth and Dietary Intake

All children had their growth assessed once they were enrolled in the study: weight was measured using a SECA (Hamburg, Germany) portable electronic baby (<10 kg) scale or SECA (Hamburg, Germany) sitting (>10 kg) scales calibrated as per hospital protocol, and length was measured using a portable recumbent length meter in children under two years of age and a fixed standing height meter in older children (rounded off to the nearest 0.1 decimal).

Food elimination was determined by a paediatric gastroenterologist on an individual basis following an allergy focused history, including a history of food eliminations trialled in the past. All children had dietary advice provided by a specialist paediatric dietitian and we used allergy diet sheets produced by the Food Allergy Specialist Group of the British Dietetic Association (https://www.bda.uk.com/, accessed on 13 May 2024) for members to support verbal information provided to the parents.

### 2.3. Assessment of Feeding Difficulties

Feeding difficulties were assessed by adapting the questionnaire used by Wright et al. [14] for the Millennium Study, which was a general paediatric population study conducted in the United Kingdom, to also include the feeding difficulties often reported in non-IgE-mediated allergies (Table 1). Parents were asked at the time of enrolment in the study to report on the presence of these feeding difficulties.

### 2.4. Ethics, Consent, and Permissions

Ethical approval for this study was obtained from the National Research Ethics Service UK (NRES London—Bloomsbury, NR: 11/LO/1177). Written consent was obtained from the parents for both participating in this study and the publication of this study.

### 2.5. Statistical Analysis

Statistical analysis was conducted in Stata (StataCorp. 2021. Stata Statistical Software: Release 17. College Station, TX, USA: StataCorp LLC). Baseline characteristics, biometric parameters, and gastrointestinal symptoms of children with and without feeding difficulties were assessed by the questionnaire and were subsequently compared. Normally distributed continuous variables were expressed as means and standard deviations (SDs) and compared between children with and without feeding difficulties with independent samples *t*-tests. Non-normally distributed continuous variables were expressed as medians and interquartile ranges (IQRs) and compared between the groups with the Wilcoxon rank sums test. The normality of distribution of continuous variables was assessed by inspection of histograms and quantile–quantile (QQ) plots. Categorical variables were expressed as absolute numbers and percentages. Univariable associations between the presence of feeding difficulties and the presence of gastrointestinal symptoms and non-IgE-mediated gastrointestinal food allergies were investigated with the Pearson chi-square test or Fisher’s exact test, as appropriate. *p*-values below 0.05 were considered statistically significant.

Logistic regression models were fit to the data to investigate significant clinical associations predictors of feeding difficulties. The initial model included age, sex, the initial questionnaire score, and the number of foods eliminated as possible predictors. Quadratic terms were tested for age and initial score to investigate non-linearity and kept in the models when significant. The final model was chosen based on backward elimination of non-significant predictors (*p*-value threshold 0.1).

## 3. Results

This study recruited 131 children, and 114 (87%) parents completed the questionnaire on feeding difficulties. The baseline and clinical characteristics and differences between children with/without feeding difficulties are summarised in Table 2. Sixty-nine per cent (90/131) were boys, and the median age was 21.8 months (IQR 7.4 to 66.2 months). The median number of foods avoided was three (IQR 2 to 4), with milk, soya, eggs, and wheat being the most common food combination, followed by milk and soya (Figure 1). Feeding difficulties were present in 61 (53.5%) of the 114 children. 

Whilst there were no significant differences for most growth parameters between the groups, children with feeding difficulties had lower height-for-age z-scores (*p* = 0.049). Additionally, these children also had a higher number of gastrointestinal symptoms (*p* = 0.007) and, based on the Likert scale questionnaire, significantly higher initial scores (*p* = 0.002). Additionally, they had overall been following an elimination diet for a longer period (*p* = 0.021).

The most common feeding difficulties were regular meal refusals (26.9%), extended mealtimes (26.7%), and problems with gagging on textured foods (26.5%) (Table 3). Most children (40/61) had ≥2 reported feeding difficulties, and eight children had ≥4. Children with feeding difficulties had higher rates of constipation and vomiting: 60.7% (37/61) vs. 35.8% (19/53), *p* = 0.008 and 63.9% (39/61) vs. 41.5% (22/53), *p* = 0.017, respectively (Table 4).

Logistic regression analysis demonstrated that the main independent predictors of feeding difficulties were age (with a quadratic relationship, as illustrated in Figure 2) and initial symptom score, while gender and the number of foods excluded in the elimination diet were not significantly associated with feeding difficulties. The adjusted odds ratios from the initial and final logistic regression models are given in Table 5.

## 4. Discussion

Whilst feeding difficulties are included in most diagnostic criteria for non-IgE-mediated food allergies, limited data are available on the prevalence of this symptom when children present with these delayed symptoms. This observational study set out to establish how prevalent it is for children with this delayed allergy to present with feeding difficulties. We found that 53.5% of parents listed at least one of the feeding difficulties outlined in our questionnaire. In fact, most of the children presented with ≥2 signs of feeding difficulties. This finding is at the higher end of the prevalence published in the recent systematic review on feeding difficulties in children with both IgE- and non-IgE-mediated allergies [9]. This is possibly due to the fact that this study only included children with non-IgE-mediated allergies, in whom feeding difficulties have more commonly been reported; moreover, the patients recruited were from a tertiary paediatric referral centre, and so are likely to represent the more severe spectrum of patients with non-IgE-mediated allergies [15]. When comparing our data to published data on healthy toddlers (30 months of age) in the UK, 20% of parents perceived their child’s eating to be a problem and 5.3% stated their child often refused to eat. The overall prevalence of feeding difficulties is therefore higher in children presenting with non-IgE-mediated food allergies and almost 27% of children regularly refused meals with this diagnosis. 

Feeding difficulties are more commonly seen in children with food allergies compared to children without food allergies [9], and our data support the clinical observations of healthcare professionals working with children with non-IgE-mediated gastrointestinal allergies. The reasons for this are likely multifactorial. It may be linked to the diagnosis itself, where gastrointestinal symptoms lead to a negative association with foods. Furthermore, the elimination diet itself limits food exposure; thus, children may also avoid foods due to the perceived danger of a possible reaction [15,16]. Maslin et al. [8] has found that the formulas used for the management of cow’s milk allergy have an impact on taste perception and may lead to an aversive eating pattern. More recently, a study by D’Auria et al. [17] on food-allergic children found impaired taste perception, which was associated with a decreased number both of the circumvallate and of the fungiform papillae on the tongue and an accompanied compositional shift in the oral microbiota, which had an impact on food neophobia when compared to healthy individuals. 

In this study, significantly more children with feeding difficulties had vomiting and constipation as symptoms and overall, a greater number of gastrointestinal symptoms. This concurs with some of the retrospective data published by Meyer et al. [10] in 2014; however, this study did not find more feeding difficulties in patients with rectal bleeding and bloating/flatus in children, compared to the retrospective study of 2014. Feeding difficulties have frequently been described in children with reflux and eosinophilic oesophagitis [11,12], and more recently, a higher prevalence of feeding difficulties was also reported in children with food protein induced enterocolitis syndrome, where the hallmark symptom is vomiting [18,19]. It has been hypothesised in children with underlying motility disorders (i.e., reflux and constipation) that repeated painful experiences (i.e., vomiting, pain on defecation) may alter the visceral sensory processing leading to increased sensory sensitivity, which in turn leads to a dysphoric feeding experience [20]. These negative experiences may have contributed to the creation of pathologic pain pathways and central pain memories. Persistent stimulation of afferent fibres reshapes the peripheral nociceptor responses as well as the central neuron responses to pain, resulting in primary and secondary hyperalgesia [21].

In this study, no significant differences were found in weight-for-age, BMI-for-age, and weight-for-height z-scores for children with and without feeding difficulties. However, children with feeding difficulties had overall significantly lower height-for-age z-scores. This may be related to the fact that this cohort of patients overall had been eliminating food allergens for a longer period, as patients were referred to our tertiary centre and had often already started elimination diets. Additionally, our data also indicated that the children with feeding difficulties presented with an increased severity of symptoms. Long-term elimination of foods, in particular cow’s milk, has been associated with stunting and increased severity of symptoms may reflect ongoing gut inflammation, which therefore impacts longitudinal growth [16,22,23,24].

Whilst our study has found significant differences between children with and without feeding difficulties, this does not prove causality. Regression analysis found that only age (with a quadratic relationship—see Figure 2) and higher initial symptom questionnaire scores were associated with developing feeding difficulties. Children go through critical oral motor and taste acceptance milestones before 1 year of age. Data show that if texture and taste expansion does not occur during this window of development, a child is at a higher risk of developing feeding difficulties [25,26]. If the early presentation was of raised severity associated with increased discomfort, this may have had an impact not only on breast or bottle feeding, but also progression with complementary foods due to a negative association with food consumption. 

This study has multiple limitations, most notably that the questionnaire used for establishing symptom improvement and establishing feeding difficulties is not a validated tool. There are validated feeding questionnaires which have been used in other studies, for example, the Montreal Children Hospital Feeding Scale [27], but none are specific to food allergies, and we were interested in capturing the feeding difficulties commonly reported and observed by healthcare professionals working with children with non-IgE-mediated allergies as well as reported in the UK. That is why we used a modified version of the feeding difficulties reported in the UK population study by Wright et al. [28]. Furthermore, one questionnaire was used for children between the ages of 7 months to 66 months, which may not capture the differences in presentation depending on age. In addition, this study relied on an elimination diet followed by reintroduction using a Likert-scale symptom score to diagnose non-IgE-mediated food allergy. The latter is dependent on parental perception and as such is a subjective measure of symptoms. The authors of this study acknowledge that a double-blind food challenge remains the gold standard to minimise bias when establishing food allergies, but this procedure is very difficult in practice, as symptoms may appear 1–4 days after exposure to the allergen. 

## 5. Conclusions

This study found that feeding difficulties are commonly reported in children with non-IgE-mediated gastrointestinal allergies and that their inclusion as a presenting symptom in children with non-IgE-mediated allergies is justified. Children with vomiting and constipation have greater feeding difficulties; in particular, the severity of symptoms in younger age seems to be associated with the development of feeding difficulties. There is a paucity in food-allergy-specific tools for establishing feeding difficulties, which requires further research in the long term and consensus in the short term amongst healthcare professions as to which tool is the best for food allergic children. 

## Figures and Tables

**Figure 1 nutrients-16-01563-f001:**
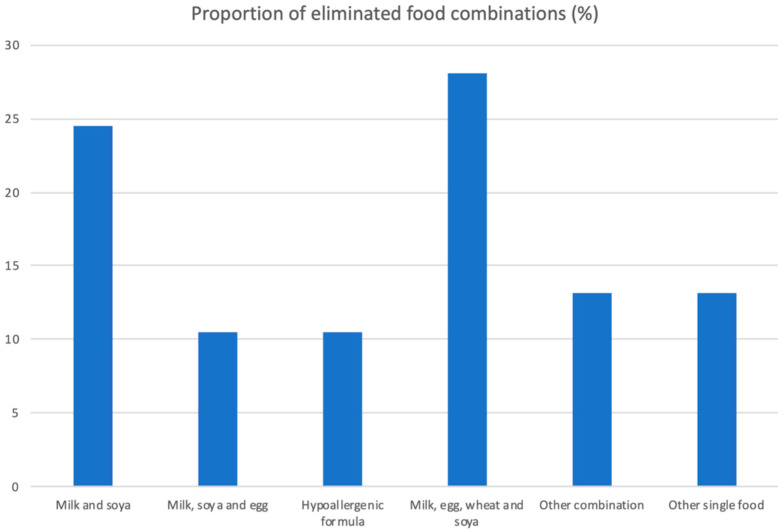
Foods eliminated by patients enrolled in this study.

**Figure 2 nutrients-16-01563-f002:**
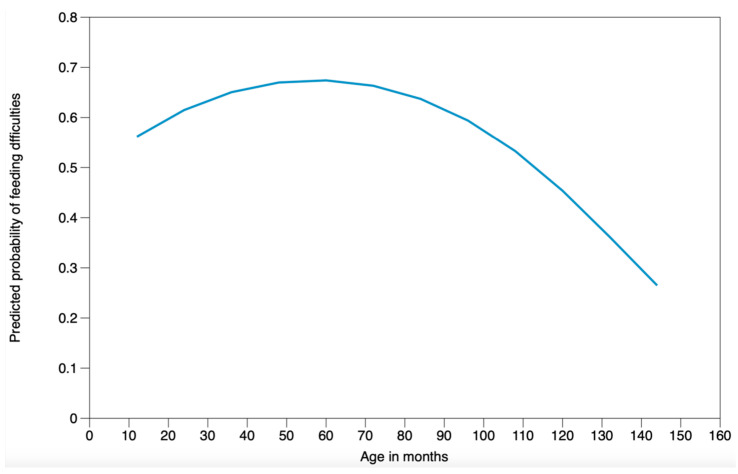
Association between age and the development of feeding difficulties. (quadratic relationship = coefficient of age × individual’s age + coefficient for initial symptoms score × individual’s symptom score).

**Table 1 nutrients-16-01563-t001:** Assessment of feeding difficulties.

Feeding Difficulties
Regular meal refusal (regular defined as daily)
Extended mealtimes (defined as >30 min)
Gagging on textured foods
Poor appetite
Dysphagia (defined as visible signs of struggling to swallow)
Difficulties with sucking (breast of bottle)

**Table 2 nutrients-16-01563-t002:** Baseline and clinical characteristics of the study population.

Characteristic	Feeding Difficulties (n = 61)	No Feeding Difficulties (n = 53)	*p*-Value	Total (n = 114)
Age (months), median (IQR)	19.4 (7.7–37.1)	24.2 (6.5–77.3)	0.428	19.76 (7.4–65.8)
Weight-for-age z-score, median (IQR)	−0.13 (−0.90–0.55)	0.13 (−0.57–0.60)	0.225	−0.01 (−0.79–0.58)
Height-for-age z-score, median (IQR)	−0.46 (−1.23–0.42)	0.25 (−0.60–0.71)	0.022	−0.27 (−0.83–0.62)
Weight-for-height z-score, median (IQR)	0.10 (−0.52–0.84)	0.43 (−0.71–0.93)	0.687	0.23 (−0.53–0.89)
BMI-for-age z-score, median (IQR)	0.04 (−0.40–0.63)	0.12 (−0.77–0.87)	0.752	0.08 (−0.53–0.63)
Gender, n (%)				
Male	37 (60.7)	39 (73.6)	0.144	76 (66.7)
Female	24 (39.3)	14 (26.4)		38 (33.3)
Number of intestinal symptoms, median (IQR)	6 (5–7)	5 (4–6)	0.007	5 (4–6)
Number of foods eliminated, median (IQR)	3 (2–4)	2 (1–4)	0.054	3 (2–4)
Symptom score before elimination diet, median (IQR)	22 (16–25)/45	17 (12–21)/45	0.002	18 (14–24)/45
Improvement, mean difference Likert scale (SD)	11.4 (7.1)	10.1 (6.5)	0.307	10.8 (6.8)
Length of elimination diet (weeks), median (IQR)	17.4 (11.7–34.8)	10.3 (7.3–21.7)	0.021	13.0 (8.7–32.6)

BMI: body mass index; SD: standard deviation; IQR: interquartile range.

**Table 3 nutrients-16-01563-t003:** Reported feeding difficulties among children with non-IgE-mediated food allergy.

Feeding Difficulties	Number/Total (%)
Regular mean refusals	29/108 (26.9)
Extended mealtimes	28/105 (26.7)
Gagging on textured foods	30/113 (26.5)
Poor appetite	21/113 (18.6)
Dysphagia	19/113 (16.8)
Breast/bottle feeding difficulties	4/113 (3.5)
Number of feeding difficulties	
0	53/114 (46.5)
1	21/114 (18.4)
2	19/114 (16.7)
3	13/114 (11.4)
4	4/114 (3.5)
5	4/114 (3.5)

**Table 4 nutrients-16-01563-t004:** Prevalence of symptoms among children with non-IgE-mediated food allergies with and without feeding difficulties.

Symptoms	Feeding Difficulties (n = 61)	No Feeding Difficulties (n = 53)	*p*-Value	Total (n = 114)
**Gastrointestinal**				
Abdominal pain	54 (88.5)	48 (90.6)	0.723	102 (89.5)
Flatus	52 (85.2)	44 (83.0)	0.745	96 (84.2)
Bloating	39 (63.9)	27 (50.9)	0.161	66 (57.9)
Diarrhoea	33 (54.1)	29 (54.7)	0.947	62 (54.4)
Vomiting	39 (63.9)	22 (41.5)	0.017	61 (53.5)
Constipation	37 (60.7)	19 (35.8)	0.008	56 (49.1)
Blood in the stool	16 (26.2)	12 (22.6)	0.657	28 (24.6)
**Co-morbidities**				
Allergic rhinitis	26 (42.6)	29/52 (55.8)	0.163	55/113 (48.7)
Asthma	31 (50.8)	29/52 (55.8)	0.599	60/113 (53.1)
Eczema	47 (77.0)	36 (67.9)	0.275	83 (72.8)
Frequent respiratory infections	2 (3.3)	1 (1.9)	1.000	3 (2.6)

Data are given in absolute numbers and percentages. Chi-square or Fisher’s exact test used for between-groups comparison.

**Table 5 nutrients-16-01563-t005:** Multivariable logistic regression estimates from initial and final models in the prediction of feeding difficulties.

Variable	Initial Model		Final Model	
	Adjusted Odds Ratio (95% CI)	*p*-Value	Adjusted Odds Ratio (95% CI)	*p*-Value
**Age in months**	1.0257 (0.9929–1.0596)	0.126	1.0271 (0.9948–1.0605)	0.101
**Age in months^2^**	0.9998 (0.9995–1.0000)	0.059	0.9998 (0.9995–1.0000)	0.047
**Male gender**	0.5871 (0.2476–1.3922)	0.227	–	–
**Initial symptom score**	1.0793 (1.0123–1.1507)	0.020	1.0873 (1.0208–1.1581)	0.009
**Number of foods excluded**	1.0715 (0.0523–3.9249)	0.504	–	–

Both models based on data from 114 children. Stepwise backward elimination used for variable selection for the final model (*p*-value threshold 0.1). The final model can be represented as: log odds of feeding difficulties = −1.629549 + 0.0267682 × Age – 0.0002332 × Age^2^ + 0.0836704 × Initial symptom score. Akaike Information Criterion (AIC): initial model 151.19, final model 149.12. Bayesian Information Criterion (BIC): initial model 167.61, final model 160.07. The background color distinguishes between models.

## Data Availability

Data are contained within the article and Appendix A.

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
