# Peer review of "Assessing Feeding Difficulties in Children Presenting with Non-IgE-Mediated Gastrointestinal Food Allergies—A Commonly Reported Problem"

_nutrients, 2024, doi:10.3390/nu16111563_

Round 1
Reviewer 1 Report
Comments and Suggestions for Authors
Assessing Feeding difficulties in children with non-IgE mediated Gastrointestinal Food Allergies. A commonly reported problem.
This publication presents the results of the latest prospective study on the relationship between non-IgE mediated food allergy in early childhood and the occurrence of functional disorders of the gastrointestinal tract, the main symptom of which is feeding difficulties.
This prospective study was performed in a prestigious pediatric center - the tertiary gastroenterology department from Great Ormond Street Hospital for Children in London. The study involved 131 children, aged 4 weeks to 16 years who met the inclusion criteria (clinical and dietary) and 114 parents who completed a questionnaire regarding their children's feeding difficulties.
Baseline characteristics of the study group, biometric parameters and gastrointestinal symptoms of children with and without feeding difficulties were assessed using a questionnaire and are summarized in Table 2.
The prevalence of symptoms (gastrointestinal and co-morbidities) among children with non-IgE mediated food allergies with and without feeding difficulties are compared in Table 4. The data obtained regarding the type of feeding difficulties, as well as the frequency occurrence of symptoms in the study group with non-IgE mediated food allergy (Tab.1, Tab.3) are similar to those previously described by Meyer et al. in 2014 as well as in children with EoE or gastroesophageal reflux (Mukkada et al., Mehta et al., and others).
Cow's milk, soy, chicken egg and wheat proteins were the main source and cause of gastrointestinal allergy at the youngest age, resulting in symptoms of feeding difficulties (Fig. 1, 2). To achieve clinical improvement, these products had to be temporarily eliminated from the diet of the study children.
Logistic regression analysis (Table 5) showed that children's age and initial symptom scores were associated with feeding difficulties as independent predictors. However, gender and the number of products eliminated from the diet did not influence the occurrence of feeding difficulties in the examined patients.
The authors should be congratulated on the discussion regarding the mechanisms of development and the type of feeding ailments in the youngest children caused by non-IgE dependent food allergy.
Lack of proper diagnosis of the cause of feeding difficulties does not allow for effective help, both for the sick child and his or her family. This is probably related to the commonly held belief that negative allergy test results exclude the occurrence of food allergy. In the case of non-IgE mediated food allergy diagnosis, the appropriate test is a diagnostic elimination-provocation test with food suspected of having a harmful effect. Physicians treating this type of disorders in young children often do not remember this.
Author Response
Thank you for this positive feedback. Based on your feedback we have made no further changes to the manuscript.
Reviewer 2 Report
Comments and Suggestions for Authors
General comment
The authors tried to find how many children had feeding difficulties in their patients with non-IgE food allergy and evaluate their clinical features by using questionnaire. Sample size was enough and the results seemed to be clear. However, the methods used in this study were not clearly described.The authors emphasized that this study was characterized as prospective research, amid there are already several reports regarding feeding difficulties in food allergy. However, it is not clear how this study was conducted in a prospective manner. Furthermore, the patients were enrolled in this study at the time when their gastrointestinal symptoms improved after 4-week elimination diet and were asked to report on the presence of feeding difficulties. It could be hard to distinguish symptoms of feeding difficulties from food allergy-associated symptoms. This might explain the higher symptom score and longer food elimination periods in children with finding difficulties compered to those without. Vomiting and constipation could be a remnant of food allergy.
Specific comments
L130: Why did the authors use the data from 131 children to show their median age and the number of avoided foods? Most findings in this study were derived from the data of 114 children. Furthermore, it was shown that the median age of 131 children was 21.8 months, but is was quite different from those of 114 children (Table 2).
Table 5: 95% CI of aOR in Age in months^2 could be 0.9995-1.0000.
Ref #13: That could be PAI 2015;26(5):403-8.
Author Response
General comment
The authors tried to find how many children had feeding difficulties in their patients with non-IgE food allergy and evaluate their clinical features by using questionnaire. Sample size was enough and the results seemed to be clear. However, the methods used in this study were not clearly described.The authors emphasized that this study was characterized as prospective research, amid there are already several reports regarding feeding difficulties in food allergy. However, it is not clear how this study was conducted in a prospective manner. Furthermore, the patients were enrolled in this study at the time when their gastrointestinal symptoms improved after 4-week elimination diet and were asked to report on the presence of feeding difficulties. It could be hard to distinguish symptoms of feeding difficulties from food allergy-associated symptoms. This might explain the higher symptom score and longer food elimination periods in children with finding difficulties compered to those without. Vomiting and constipation could be a remnant of food allergy.
Thank you for this comment. As mentioned in the text, the methodology has been published in more details. We have added further details to allow readers to understand how patients were recruited.
In regard to the distinguishing of feeding difficulties from symptoms of non-IgE mediated allergies. Feeding difficulties is a presenting symptom for non-IgE mediated allergy (but there is little data to support this inclusion in our guidelines, as per our introduction) and distinct from other presenting symptoms, so it is easy to distinguish between the two. This study set out to justify the inclusion of feeding difficulties as a diagnostic criterion by establishing prevalence at presentation and also assessing improvements. We do acknowledge that this could be made clearer for the reader, so we have suggested a change in the title and we have also made a minor amendment to our objectives to make this clearer.
Specific comments
L130: Why did the authors use the data from 131 children to show their median age and the number of avoided foods? Most findings in this study were derived from the data of 114 children. Furthermore, it was shown that the median age of 131 children was 21.8 months, but is was quite different from those of 114 children (Table 2).
Thank you for this comment. We have changed this to 113 children and changed all of the statistical analysis.
Table 5: 95% CI of aOR in Age in months^2 could be 0.9995-1.0000.
Thank you, this has been corrected.
Ref #13: That could be PAI 2015;26(5):403-8.
Yes this is.
Reviewer 3 Report
Comments and Suggestions for Authors
The subjects included are of a vast variety of pediatric ages, comprising at least 4 age groups with quite different psychological and physiological characteristics. I think it is important to separate them and describe their age-related characteristics. The severity of reactions is also very important - about 40 % of anaphylaxis is also non-IgE-mediated. I think that it is also important to investigate the reason or the origin of the feeding problems - because of the child, the parents, grandparents, caregivers, kindergarten, and friends. I think that the problems with food preference/feeding depend quite a lot on the ability of parents to prepare dietetic food and to present it to the child in an attractive way. I think also that it would be fine to have a comparison between children with IgE-mediated allergies and those without any diseases.
Author Response
The subjects included are of a vast variety of pediatric ages, comprising at least 4 age groups with quite different psychological and physiological characteristics. I think it is important to separate them and describe their age-related characteristics.
Whilst this is a very valid point, we did not set out to describe the characteristics at different ages, but rather the prevalence of feeding difficulties as a presentation symptom, to justify the inclusion of this symptoms as a presenting symptom for non-IgE mediated allergy. We have included this as a possible limitation of the study.
The severity of reactions is also very important - about 40 % of anaphylaxis is also non-IgE-mediated.
Thank you for this comment. We had no patients that had any anaphylaxis in our study. Whilst there were some that had IgE sensitisation, these patients did not have anaphylaxis. The data on 40% of anaphylaxis being non-IgE mediated is related to drug allergies and not non-IgE mediated food allergies. https://pubmed.ncbi.nlm.nih.gov/28590439/#:~:text=Approximately%2060%25%20of%20perioperative%20anaphylactic,type%20B%20adverse%20drug%20reactions).
Whilst this is a very important point, the severity of reaction like you would judge IgE mediated allergies was not applicable. Our study did not include children with FPIES, which would be the severest form of non-IgE mediated allergic children.
I think that it is also important to investigate the reason or the origin of the feeding problems - because of the child, the parents, grandparents, caregivers, kindergarten, and friends. I think that the problems with food preference/feeding depend quite a lot on the ability of parents to prepare dietetic food and to present it to the child in an attractive way. I think also that it would be fine to have a comparison between children with IgE-mediated allergies and those without any diseases.
The origin of feeding difficulties are quite well studied in paediatrics and there are many factors that the reviewer is correct that influence feeding. However, in the case of non-IgE mediated allergies, feeding difficulties is primarily driven by the organic disease (pain, discomfort) that parents respond to and this is part of the presenting symptoms. This is described by Chehade et al. in a publication with one of the authors of this paper. https://pubmed.ncbi.nlm.nih.gov/30922955/
This publication aimed at justifying the inclusion of feeding difficulties as presenting symptom for non-IgE mediated allergies.
Thank you for this useful thought on a comparison study. There are currently no age matched data bases for feeding difficulties in IgE mediated food allergies that we could use. In fact, there is paucity of data in IgE mediated allergy. We just published our systematic review finding on this.
https://pubmed.ncbi.nlm.nih.gov/38566436/
This would be a welcome new study on IgE mediated allergies but was not included in our study as the hospital where the data was generated specialises in non-IgE mediated allergies.
We have however added a comparison to the healthy UK population to the text, which would be useful to compare to the levels feeding difficulties in the non-IgE mediated population.
Round 2
Reviewer 2 Report
Comments and Suggestions for Authors
By the authors’ revising the manuscript, the methods became clearer than before. Now I understand the reason why only the children whose GI symptoms were improved after a 4-week elimination diet were included in the analysis was to make a strict diagnosis of non-IgE mediated food allergy. If so, it would be preferred to describe the criteria for significant improvement in GI symptoms by the elimination diet.
I agree that there were few recall biases about feeding difficulties because the data were obtained at the time of enrollment. But still I can’t understand why this study was categorized as a prospective one. The 4-week follow-up was only done for making sure that the diagnosis was correct. When the authors compared the degree of feeding difficulties before and after the elimination diet, the study would become prospective.
Author Response
By the authors’ revising the manuscript, the methods became clearer than before. Now I understand the reason why only the children whose GI symptoms were improved after a 4-week elimination diet were included in the analysis was to make a strict diagnosis of non-IgE mediated food allergy. If so, it would be preferred to describe the criteria for significant improvement in GI symptoms by the elimination diet.
Thank you for this useful comment. This has been extensively described in our publication, which is open access (https://onlinelibrary.wiley.com/doi/10.1111/pai.12404). In children with Non-IgE mediated allergies there is no standardised diagnostic tool (even the COMISS score is not diagnostic but an awareness tool and not specific to non-IgE mediated allergies. As every child starts with a different score and is reliant on the perceived severity it makes it impossible to have a cut-off. Therefore, every child is their own control to establish improvement. We have added a sentence to explain this better. Furthermore we have added this again to the limitation as the original study.
I agree that there were few recall biases about feeding difficulties because the data were obtained at the time of enrollment. But still I can’t understand why this study was categorized as a prospective one. The 4-week follow-up was only done for making sure that the diagnosis was correct. When the authors compared the degree of feeding difficulties before and after the elimination diet, the study would become prospective.
Thank you – we have changed this to just an observational study